# Dough Rheological Properties and Macronutrient Bioavailability of Cereal Products Fortified through Legume Proteins

Chaima Neji [1], Jyoti Semwal [1,2], Endre Máthé [1] and Péter Sipos [1,*]

1    Faculty of Agricultural and Food Sciences and Environmental Management, Institute of Nutrition, University of Debrecen, Böszörményi út 138, 4032 Debrecen, Hungary
2    Department of Grain Science and Technology, CSIR-Central Food Technological Research Institute, Mysore 570020, Karnataka, India
*    Correspondence: siposp@agr.unideb.hu

**Abstract:** Cereal products are regarded as important protein providers, though they could feature poor nutritional quality due to their occasional imbalanced amino acid content. Cereal proteins are low in cysteine or tryptophan, and rich in methionine; however, while their combination with legume proteins makes them nutritionally more comprehensive, such a possibility must be addressed by the cereal processing industry. However, the incorporation of legume protein concentrates and isolates might also influence the functionality and bioavailability of some cereal constituents. Therefore, the objective of the present review is to gain insights into the effects of cereal products incorporated with legume protein isolates/concentrates, knowing that both the cereals and the protein extracts/isolates are complex structural matrices, and besides the final products acceptability they should efficiently promote the health condition of consumers. The combination of legume proteins with cereals will bring about a structural complexity that must harmoniously include proteins, carbohydrates, lipids, polyphenols and dietary fibers to promote the bioaccessibility, bioavailability and bioactivity without cyto- and genotoxicity.

**Keywords:** legume protein; cereal products; rheology; interactions between nutrients



## 1. Introduction

Nowadays, legume protein isolate/concentrate application is diversifying the food industry so that they are used as meat replacers/analogs, plant-based milk substitutes and gluten-free bakery products [1] (Figure 1). One of the main reasons behind the growing utilization of legume proteins is the improvement of the nutritional and functional characteristics of food products [2,3]. Therefore, the analysis of the effectiveness of these proteins in influencing the sensory and physicochemical properties of fortified foodstuff, together with the obtained texture and nutritional features, are to be addressed [3].

Cereals are used extensively for several staple productions, such as bread, pasta, biscuit, cake and a wide range of snack foods [4]. Nevertheless, cereal products are also criticized for their limiting lysine content, while a high amount of this essential amino acid is present in legumes, such as peas and beans. On the other hand, cereals contain a relatively high amount of methionine, which is present in low amounts in legumes, thus, the combination of cereals and legumes makes them nutritionally complementary [5,6]. Methionine is not synthesized de novo in humans/animals cells, and usually represents the first amino acid included into the polypeptide chain during protein synthesis. Additionally with the rising incidence of gluten allergy among population, there is a growing demand for gluten-free bakery products [7]. Since most gluten-free products have lower protein content than their counterparts [8], the addition of legume protein isolate could not only be a good option for fortification, but also a way to develop protein rich gluten-free products.

However, during the development of food products, the interaction of protein with other ingredients such as carbohydrates and phenols, can affect the product functionality [9]. Therefore, understanding the effects of processing on the behavior of legume proteins within the food matrix is an essential feature of every newly developed cereal-legume-based foodstuff. In particular, the characterization of the molecular interactions that can be formed between legume proteins and other food ingredients are to be considered in every such foodstuff, and consequently must define/apply the most suitable manufacturing technology. The different processing methods can have significant impacts on the structural, techno-functional and nutritional properties of protein isolates/concentrates, as was reviewed in our previous paper [10]; nevertheless, the specific behavioral pattern of the newly developed food matrixes should also be carefully addressed. The addition of legume proteins to cereal products can have a major impact not only on the nutritional, but also on the functional characteristics of the fortified foodstuff. The objective of the present review is to gain insight into the nutritional aspects of cereal products that were fortified through the incorporation of legume proteins, and how this fortification influences the rheological properties of the different types of dough during food processing.

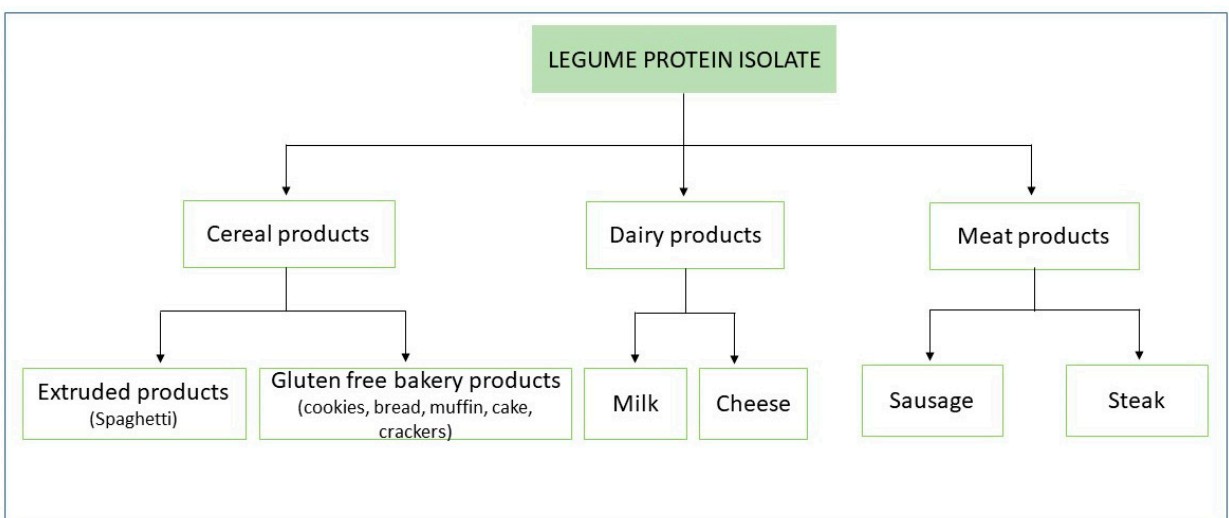

**Figure 1.** Application of legume protein isolates for food fortification.

## 2. Effect of the Legume Protein Isolate/Concentrate on the Rheological Properties of Raw Material

The evaluation of the rheological parameters in food industrial practices is crucial [11]. The texture mainly depends on the viscoelastic properties of the ingredients, especially those that showing both individually or in combination both elastic and viscous features [12], and thus affect the molding characteristics and the quality of the end products [13,14]. For instance, a viscoelastic dough/batter is necessary to entrap air and gases during heating in order to achieve the desired product volumes [15,16]. One of the methods to determine viscoelasticity of the raw material is a dynamic measurement that provides simultaneous information on both the elasticity and viscosity properties [17]. The viscoelasticity is illustrated in terms of storage modulus (elastic behavior, $G'$), loss modulus (viscous behavior, $G''$) and loss or damping factor (tan $\delta = G''/G'$) [18,19].

Studies showed that the viscosity of proteins during processing can change, thereby altering the rheological properties of the food product [12,18]. Therefore, a better understanding of rheological properties could substantially advance the development of high-quality and improved protein containing products, by controlling the manufacturing technology and modulating the product properties [18,20]. Many researches addressed the rheological properties of legume protein isolates such as faba bean [21], chickpea [22], kidney bean and field pea [23]. Studies also showed that the rheological properties of proteins, isolated from different sources seemed to depend on the β-sheet content in protein

secondary structures [23]. Furthermore, food processing techniques and conditions during product development, such as extrusion (hot or cold extrusion and/or mechanical stresses) and the variation of pH and/or thermal treatments can influence the viscoelastic properties of legume proteins isolates (such as soybean, carob and mung bean protein) [19,20,24–29]. Accordingly, many studies did report the modulation of viscoelastic properties of soybean, cowpea and pea protein isolates by high pressure processing and heat treatment [30–32]. Moreover, the effect of microwave irradiation on the viscoelastic property and microstructure of soy protein isolate derived gel were also studied [33].

## 2.1. Dough Rheology
### 2.1.1. Bread

During bread formulation, the addition of soy protein to wheat flour seemed to interfere with gluten, and this indirection could be of direct or indirect type due to the presence of water [34]. Such interaction leads to protein aggregation and high water retaining capacity that also increases elasticity (G′ increased and G″ and tan δ decreased) and stability of wheat dough [35]. Accordingly, the increased elasticity of carob, pea, lupin and faba bean proteins was also observed, which was supported by the observed low damping factor [18]. The incorporation of soybean protein to wheat flour increased the disulphide linkage, and improved elasticity, but it resulted in weak gel formation and poor heat-setting capacity, and as a consequence low quality bread was obtained [35]. Moreover, such an interaction also decreased the gas retention capacity of the dough in comparison with wheat dough, and led to the formation of a gluten network with high carbon dioxide permeability [34]. Likewise, the addition of lupin protein reduced the resistance of dough to prolonged kneading [36], and conversely, it increased the dough developmental time together with stability along with the deformation resistance and dough extensibility [37].

As mentioned above, the incorporation of legume protein can influence the water absorption capacity of the bread dough that could affect the distribution of the dough components hydration and the gluten network development [37]. The supplementation of wheat flour with pea protein isolate/concentrate and soybean concentrate also increased the dough water absorption. This was attributed to the high-water absorption capacity of the corresponding proteins that were limiting the necessary water for the development of the optimal gluten network [37–39]. Furthermore, the amino acid composition of the added proteins could also be a limiting factor regarding the water absorption capacity of dough [37]. Interestingly, the lupin protein being rich in globulins as compared to wheat flour did increase the necessary water content for optimum bread making, while this parameter remained unaffected when lupine protein rich in albumin subunit was added. This variation was attributed to the highly water-soluble albumins, which require less water to hydrate thoroughly [37]. However, the water absorption of bread dough, containing thermally denatured cowpea protein, was significantly enhanced resulting in soft texture [40]. In addition, the incorporation of 3% soybean protein isolate or an extruded soybean protein isolate resulted in a continuous and dense gluten network similar to that of only wheat flour made [41]. The above-presented observations are indicating that the decrease in gluten content results in more elastic bread dough with altered water requirement.

Apart from this, the attempts to develop gluten-free bread for celiac patients showed that the removal of gluten from bakery products impairs ability of dough to behave properly during leavening and baking [42]. Therefore, the incorporation of protein isolate/concentrate to replace the gluten may enhance the dough specific rheological performance and the technological quality of breads. The G′ of pea isolate, lupine (59% protein), soybean protein concentrates were higher than those of loss modulus (G′ > G″) that indicates dominant elastic properties [43]. This predominant elastic behavior was imparted to rice dough on the addition of pea and soybean protein isolate [44]. In case of corn and potato starch dough, the incorporation of soybean protein exhibited insignificant effect on the values of G″, whereas it caused a visible increase of G′ [45]. This effect signifies strengthening of elastic structure of the dough, despite of its comparable ability to dissipate

energy [45]. Furthermore, the extent of the effect of protein isolates greatly depend on the nature of the proteins [44]. For instance, soybean protein resulted in higher increase in G′ and G″ with a noticeable reduction in phase shift tan δ along with loss of the relation between moduli and oscillation frequency when compared to pea and lupine proteins [43] Similar decrease was also found in the tan δ when soybean protein was added to starch [15]. On the contrary, an increase in tan δ was recorded after the supplementation of rice cassava dough with soybean protein [46]. However, the use of lupin protein in dough formulation did not affect the value of the phase shift tangent [43]. The mixing of pea protein isolate with starch did not result in significant differences regarding the storage modules of the dough, while it exhibited higher tan δ value as compared to carob and lupin proteins that feature high viscous property [15]. A comparison of the rheological characteristics of dough and batter enriched with legume protein extracts is shown in Figure 2. It can be observed that the enrichment resulted rather elastic character in dough and butter than viscous, since each sample is located in the diagram space below δ = 45° based on the G′ and G″ values. Regardless of the magnitude of the values, the loss factor is normally around 0.32, which represents a phase shift of about 18°. Since the values of tan δ varied in the range 0.1 < tan δ < 0.5, the formed systems are qualified as weak gels [43].

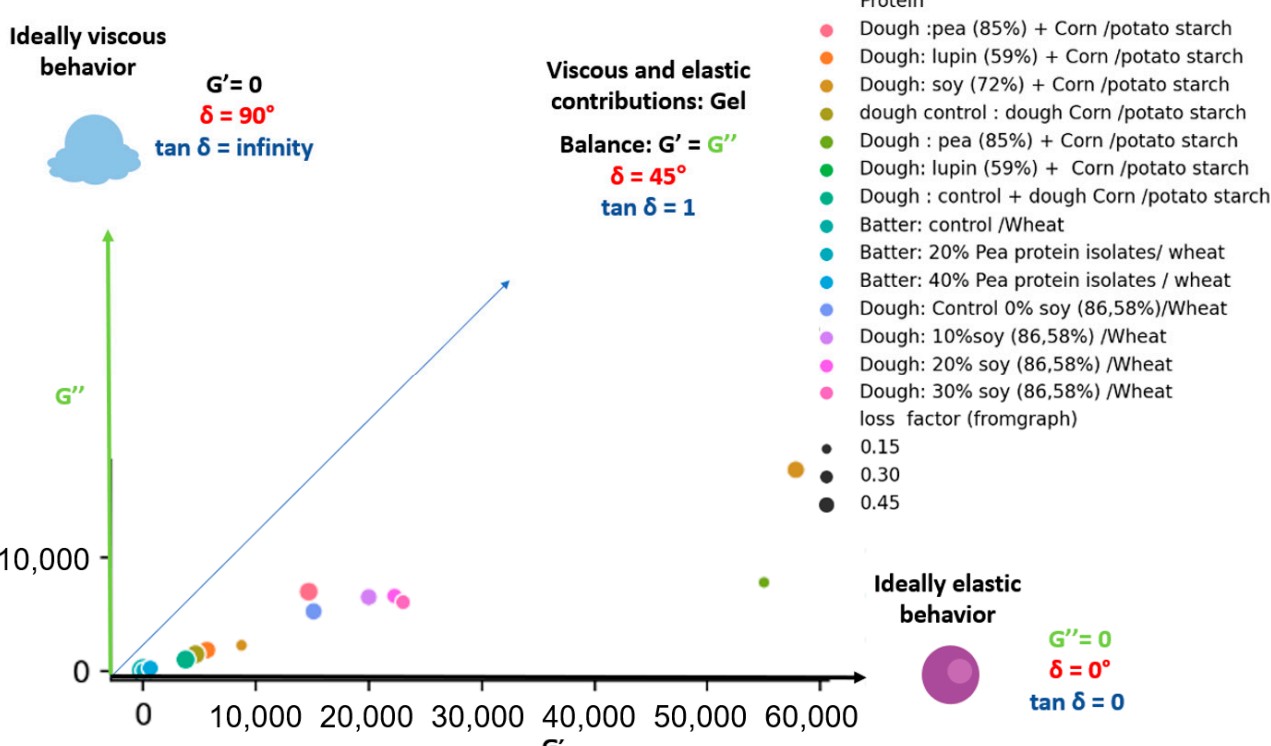

**Figure 2.** Effect of the incorporation of legume protein isolate on G′, G″ and tan δ (frequency: 1 rad/s) [35,43,45,47].

### 2.1.2. Pasta Noodle, Pasta and Spaghetti

For noodles, the addition of chickpea protein to rice dough resulted in weak gel-like property with higher G′ than G″. The high G′ of noodle samples was associated with the strong interaction between chickpea protein and rice starch due to the stabilized dough network [48]. Similarly, the addition of native and texturized soybean protein isolate to the wheat dough gave higher G′ and G″, as compared to only wheat dough. However, the fortification of wheat flour with texturized soybean protein produced a dough with more solid-like properties [49].

The changes in rheological features seem to depend upon the amount of added protein isolate. Accordingly, the addition of white lupin protein up to 20% resulted in less extensible

doughs. However, such a dough became weak when the protein content was increased to 50%, and similarly it decreased stability, development time, extensibility and resistance [50]. During the extrusion process, the type of the screw extruder can also affect the rheological characteristics of dough and product. The use of the single-screw for 0%, 5%, 17% and 30% lupin protein isolate gave a more cohesive structure than twin-screw extrusion [51]. Additionally, the single-screw extrusion process performed better in preserving the textural and cooking properties commonly accepted for pasta [51].

## 2.2. Batter Rheology
Cake and Muffin Batter

Studies showed that the batter viscosity can be affected by a number of variables, such as composition of raw material (protein, starch and pentosan), particle size, the amount of water, solids concentration, other additives and their interactions, and processing conditions, such as temperature [17]. However, the rheological properties of the batter and the technological characteristics (specific volume, color, and texture) of the muffin are determined by the type of protein used in the formulation of manufacturing recipes [52].

The addition of legume protein can improve the batter consistency and hence, it can impart superior quality to the final product. The addition of soybean protein increased the protein content in wheat flour batter along with its consistency, which was partially attributed to the intramolecular bonding and intermolecular interactions with gluten [53]. On the contrary, the addition of lentil protein did not affect rheological properties of the ingredient mixture, however, it contributed in crumb structure strengthening and enhanced entangled network in both cake and muffin [54].

Generally, the presence of protein decreases tan δ [55]. In comparison to wheat protein, the decrease in tan δ was high when 10 and 20% of soybean protein isolate was added to rice, corn, potato and wheat starch, which became evident with the increase in concentration of soybean protein [55]. Similarly, the addition of kidney bean and field pea protein to prepare starch-based batter decreased tan δ but increased $G'$ and $G''$, which in turn enhanced batter viscoelasticity [56]. Therefore, the specific volume, springiness and cohesiveness of muffins increased. However, the firmness of the muffins varied with the source of protein isolate [56]. The addition of soybean protein isolate to rice starch gave higher consistency, adhesive force, $G'$ and $G''$, which exhibited similar rheological properties to that of wheat flour batters [55]. Similarly, the addition of pea and soybean protein isolate to rice-based muffin batter did increase the $G'$, which further augmented the temperature growth [52].

It is widely accepted that the increase in dynamic moduli is related to the availability of free water that plays a critical role in modulating the viscosity since the starch granules could not dissolve in cold water [57]. Hence, the presence of proteins might reduce the availability of free water due to their high water absorption capacity. Since the free water facilitates the particle movement in the batters, its reduction could increase the batter viscoelasticity [56]. In the same context, the increase in dynamic moduli could be attributed to high water absorption capacity (WAC) of protein maize blend having reduced amount of free water available [58]. According to [59], the fortification of rice flour with red cowpea protein isolate did improve the viscoelasticity of the batters due to an increased capacity to bind/absorb water. Similarly, the case of the increase in wheat batter consistency could also be invoked the high water absorption capacity of the added soybean protein [53].

## 3. Effect of Legume Protein Isolate/Concentrate on Digestibility and Nutritional Quality of Cereal Foodstuff

The plant-based food system predominantly comprises proteins and polysaccharides such as starch, cellulose, glucomannan, pectin, hemicellulose, gums and mucilage [60,61]. Therefore, the study of molecular interaction in a food system is of an utmost importance in the case of fortifications as these interactions might also affect the inclusion of external proteins in the newly developed foodstuff matrixes [62] (Figure 3).

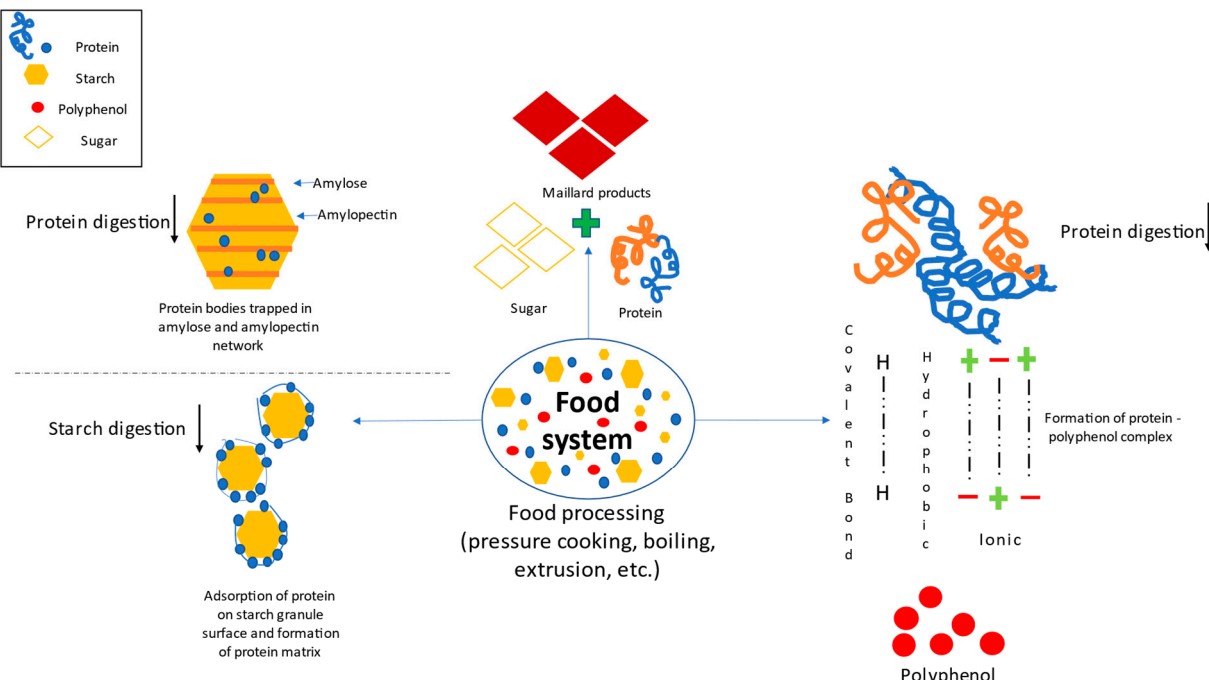

**Figure 3.** Protein-polyphenol, protein-starch and protein-sugar interaction.

*3.1. The Protein-Starch Type of Interaction and Their Relevance*

Among polysaccharides, starch is the major storage carbohydrate of plants [63], which provides a large interface for starch-protein interaction. Hence, it is important to understand the interactional effects on each moiety because ultimately such interaction should facilitate the bioaccessibility, bioavailability and bioactivity of nutrients. We define bioaccessibility as the amount of ingested and digested food ready for absorption in the intestinal tract, while bioavailability refers to the ingested and further processed nutrient fraction that would reach its ultimate cellular target(s) and bring about its specific biological function(s).

3.1.1. Effect of Protein on Starch Digestibility

The starch-protein interactions can influence their digestibility irrespective of the source of origin. The fortification of wheat noodles with heat treated pea protein isolate (at 85 °C for 30 min) reduced significantly the rate of glucose release in in vitro conditions when compared with native pea protein isolate. The authors attributed such an effect to the reduced degree of gelatinization as a consequence of proteins binding to starch [64]. Similarly, the addition of extruded lupin protein isolate (17% with twin-screw extrusion and 30% with single-screw) to spaghetti reduced the starch digestibility owing to the coating mechanism of protein matrix over the starch granules [51]. Research confirmed that the thermal and pressure processing of soybean protein isolate significantly influenced its binding kinetics. The study revealed that the adsorption of treated proteins over the starch granule surface was higher than in case of untreated proteins [65]. The faba bean protein-based fortification of the wheat crackers had no relevant effect on its starch digestibility [66]. In contrast, the effect of cooked pea and soybean protein isolate (at 100 °C for 30 min, 15 psi) seemed to diminish the rapidly digestible starch content of wheat starch [67]. On the other hand, the use of soybean protein in combination with corn starch did increase the resistant starch content and decreased the rapidly and the slowly digestible ones [68]. However, the extruded pea protein supplementation did not influence the release of soluble glucans and glucose in wheat starch unlike the hydrolyzed pea protein that significantly reduced starch amylolysis [69]. In the case of semolina-based pasta production, the addition of 25% of native faba bean protein isolate and concentrate increased the total dietary fiber and slowly digestible starch content. The authors added that the addition of faba bean flour

and starch isolate slightly increased protein and starch concentration with alterations in the amylopectin architecture and the amylopectin fingerprint region, which seemed to behave like a kind of hybrid matter between cereal and legume starch [70].

The occurrence of protein bodies over the starch surface seemed to anchor more binding proteins, and maintain this interaction against desorption, thereby will hinder starch hydrolysis [65,68]. On the other hand, the formation of three-dimensional network with different pore size during cooking might be another possible explanation for the reduction of the starch enzymatic digestion. In particular, the small pore size of starch network comparing to soybean protein concentrate network, indicate that starch exhibited a more rigid structure [70].

Moreover, the starch-protein interactions can be further enhanced by physical treatments. For instance, extruded samples showed improved starch-protein interactions, where the magnitude of interaction was found to be strongest in the blend with denaturized and/or hydrolyzed proteins via hydrogen bonding [67,69]. Furthermore, researches identified an interaction mechanism between starch-protein type of mixes, which is emphasized by the encapsulation of starch granules with proteins that obstructs the amyloglucosidase action [67]. The protein matrix can also act as a barrier towards starch digestibility, which might be substantially strengthened by protein denaturation and the applied cooking processes, such as pressure cooking and boiling.

### 3.1.2. Effects of Starch on Proteins Digestibility

Right after food enters the digestive system, the amylose and amylopectin [71] components of starch could hinder the digestion of other proteins [72]. The fortification of basmati rice starch, containing 20–25% amylose, with pea protein resulted in a diminished protein digestibility when compared with a glutinous starch containing only 0–3% amylose. This decrease was associated with the integration of the proteins into the amylose network that was formed after the leaching of amylose. Accordingly, the low amylose type of starch promotes the formation of a less extensive amylose network that increases the digestibility of proteins [72]. Interestingly, the pea protein-based meat substitutes exhibited higher protein digestibility when the amylose content was increased [73]. Apart from that, the addition of pure amylopectin also decreased digestibility of pea protein extrudate. The amylopectin could improve the flexibility of pea protein molecules and promoted the aggregation of proteins, thereby decreasing the in vitro protein digestibility [73]. Furthermore, the Maillard reaction between the degradants of amylopectin and pea proteins could reduce the sensitivity of digestive proteases, and ultimately decreasing the bioavailability of pea protein [73]. It is also true that we are missing the noninvasive and direct assessment enabling investigation type of real time methods when it comes about the exact amino acid (both essential and non-essential) content of different pools existing at the level of specific organs, tissues and cells.

### 3.2. Effects of Protein Combinations on Their Digestibility

Other than starch, the addition of external protein (of the same or from different source) could interact with native protein moiety affecting its digestibility. A study on the effect of durum wheat based semolina fortification with chickpea flour/protein isolate revealed that unlike chickpea flour, the protein isolate could decrease the protein digestibility [74]. Moreover, it was observed that as the protein isolate proportion increased concomitantly the protein digestibility was diminishing. To explain such an effect, it was stipulated that the added proteins could facilitate a high number of covalent bonds between the protein bodies, and produced a network with low susceptibility towards protein hydrolysis [74]. Quite exceptionally, the combination of milk protein concentrate, with plant protein isolates (i.e., soybean, rice and pea), increased digestibility [75]. Additionally, these blends of proteins from different sources showed better antioxidant activity than the individual protein isolates [75]. The above-mentioned studies suggests that the protein digestibility depends heavily on the source of the mixed protein (animal or plant origin) and their concentrations.

### 3.3. Effect of Polyphenols on Protein Digestibility

Plant polyphenols display natural binding affinity for proteins [76]. Their interactions occur naturally in most foodstuff and may affect their bioavailability, bio-accessibility, and bioactivity that would depend on the chemical structure of phenolic compounds and their interacting proteins [76]. It is also believed that a better understanding of the mechanisms for such interactions could improve food processing conditions to facilitate the maximal health promoting effects of polyphenols.

The type of polyphenol (flavonoid and non-flavonoid) is one of the parameters that can affect protein digestibility [77]. For instance, pea proteins in carrot puree were more digestible than in apple puree, which was ascribed to the presence of procyanidins in the apple [78]. Another study demonstrated that cranberry polyphenols could bind pea protein isolate, slowing down its in vitro digestion rates by approximately 25% in gastric (pepsin) digestion and 35% in intestinal (pancreatin) digestion [79]. Likewise, the derivatization of soybean proteins with chlorogenic acid and quercetin induced a decrease in the amount of free amino, thiol groups and tryptophan, resulting in a change in the digestion behavior [80]. Epigallocatechin-3-gallate, chlorogenic acid and resveratrol had different effects on pea protein isolate, which was related to the conformational changes of protein and polyphenols after binding. These changes increased the susceptibility of protein isolates towards enzymatic hydrolysis [81]. The complexation of soybean protein isolate with anthocyanin-rich black rice extracts, improved the digestibility of the complex. The improvement of the rate of protein hydrolysis in the complex was associated with the formation of a soybean protein isolate-anthocyanin-rich black rice extracts network that promoted an enzymatic action on the protein isolate [82]. More than that, the concentration of polyphenols [77] and the type of protein-polyphenol interaction can affect the protein digestibility. A study showed that non-covalent complex of soybean protein and epigallocatechin gallate is more digestible than the covalent complex.

Research data are indicating that in some foodstuff the higher the concentration of polyphenols the lower will become the protein digestibility. The high concentration of epigallocatechin gallate in a protein-polyphenol complex was shown to reduce the protein digestibility [83]. The protein-polyphenol complex formation and their interactions are also influenced by the applied food processing methods and the food pH [77]. At pH 2.0 and 4.6, pea protein displayed high degree of interaction with blueberry polyphenols than at pH 6.8 and 7.4 [62]. However, the evaluation of the free amino acid content of the protein isolates before and after digestion in the presence and absence of blueberry polyphenol revealed that complexation did not affect the digestion of any of the proteins [62].

The higher release of hydroxycinnamic and chlorogenic acids (CHAs) during the proteolytic digestion of soybean protein isolate-CHA, compared to egg white-CHA and whey protein isolate-CHA, was related to the smallest CHA bindings. It is supposed that bound CHAs probably decreased the availability of peptide bonds for proteolytic enzymes. Consequently, the protein digestion became more complex, preventing the release of hydroxycinnamic and chlorogenic acids from hydrophobic bonds [84]. Although the molecular weights and the amino acid profiles of soybean protein isolate and conditions (such as pH and denaturation temperature) affected the soybean protein—CHAs interaction, they did not affect the amount of the released CHAs during proteolytic digestion [84]. Besides the effect of protein-polyphenol interaction on protein digestibility, their complexation can provide health benefits. The non-covalent complex of protein and polyphenol exhibits better nutritional value and hence, it can be used as a functional food ingredient. In contrast, the covalent complex has higher stability, protects polyphenols from decomposition, and therefore, it is more suitable as an active material transport carrier or biological material [83]. The interaction between grape polyphenol—soybean protein isolate can retain and possibly amplifies the health benefits of polyphenols. The uptake of a single dose of 300 mg/kg or 500 mg/kg of grape polyphenol—soybean protein isolate complex, having 5% grape polyphenols significantly lowered blood glucose in obese and hyperglycemic C57BL/6 mice 6 h after administration [85]. Peptides from peanut protein (50%

protein)–cranberry polyphenol complexes and peanut protein–green tea polyphenol complexes were substantially less immunoreactive compared to peptides from uncomplexed peanut flour [86]. Moreover, the interaction between protein-polyphenol is considered as a promising approach to improving the antioxidant activity of proteins [77]. For example, the covalent cross-linking of soybean protein isolate with tannic acid in an alkaline environment improved its antioxidant activity [87]. Likewise, the combination of soybean protein isolate and grape seed procyanidins effectively enhanced the antioxidant effects of the grape seed procyanidins [88]. After storage, the antioxidant properties of grape seed procyanidins-soybean protein isolate containing complex solutions were showing higher values than the grape seed procyanidin solutions. This result was explained by the effect of soybean protein isolate on the embedding and controlled release of grape seed procyanidins that got reduced [88]. The above-discussed studies suggested that protein-polyphenol complex could improve the antioxidant effects of polyphenols along with alleviation of allergenic characteristics of proteins. Therefore, such complexes might find an application in food industries as an important health-promoting functional ingredient. Furthermore, the above-discussed studies confirms that the protein digestibility is greatly influenced by the polyphenols specific source, type, concentration and interaction mechanisms with proteins. Furthermore, a detailed study on the protein-polyphenol type of interactions seems essential before embarking on the fortification of any food system with proteins [77].

*3.4. Effects of Protein Fortification on Lipid Digestibility*

During digestion, the lipid-protein type of interactions significantly affect lipolysis [89]. In order to evaluate the extent of lipolysis, the triglyceride transformation is monitored as one of the relevant parameters. The latest it represents the proportion of triglyceride that have undergone hydrolysis in relation to the intact triglyceride that was initially present in the analyzed serum sample [90]. In addition to the triglycerides, the total cholesterol, low-density lipoprotein cholesterol and high-density lipoprotein cholesterol are considered as important lipid fractions and markers that attract increased clinical attention [91]. The above mentioned lipid markers specific concentrations are also related to the development of atherosclerotic cardiovascular disease, obesity, inflammation and metabolic syndrome [92]. Research observations did demonstrate that the addition of soybean protein isolate to a slightly oxidized sunflower and flaxseed oil could induce and improve the hydrolysis of triglycerides, compared to diglycerides and monoglycerides during an in vitro digestion experiment [90]. A lower cholesterol and triglyceride activity was noticed after the addition of pea [93] and lupin protein isolates [94,95]. Similarly, the consumption of 25 g of lupin protein isolate can beneficially modulate plasma LDL cholesterol at least over short period of time [96].

Another study performed among a type 2 diabetic patients with nephropathy, showed that a 4-year long soybean protein substitution (0.8 g protein/kg body) in the diet resulted into a significantly lower levels of total cholesterol, LDL cholesterol, and triglycerides [97]. Furthermore, the incorporation of isolated soybean protein led to a decreased LDL cholesterol concentration (3%), but without significantly affecting the HDL cholesterol, triglycerides, lipoprotein or blood pressure [98].

To understand the mechanism behind these effects, in vitro and in vivo studies were performed. The cholesterol reduction was associated with stimulation of LDL receptors by a well-defined protein component (conglutin) in the case of lupin protein isolate [94]. For the same protein isolate, the hypotriglyceridemic effect was related to the downregulation of sterol regulatory element-binding protein (SREBP-1c) encoding gene in the liver, which reduced the hepatic fatty acid synthesis [95]. On the other hand, ref. [93] claimed that pea proteins affected cellular lipid homeostasis by upregulating genes involved in hepatic cholesterol uptake and downregulating fatty acid synthesis genes [93].

Many factors can affect the lipid hydrolysis during the gastrointestinal digestion phase. Food bolus composition is one of the factors that impacts lipid hydrolysis and oxidation reactions [90]. In particular, as one of the food components, protein can significantly

influence the extent of lipid oxidation based on their nature [90]. For example, the presence of non-adsorbed proteins in the aqueous phase is considered the most crucial factor affecting the rate of lipid oxidation. They inhibited lipid oxidation by binding transition metals and reducing their ability to interact with ω-3 fatty acids in the lipid droplets [99]. On the other hand, the antioxidant properties of the released amino acids/peptides could also affect the extent of lipid oxidation and the reactions pathways [90] since many protein antioxidant mechanisms are dependent on their amino acid composition (e.g., metal chelation, free radical scavenging, hydroperoxide reduction, aldehyde adduction) [100]. Studies showed that the lentil, pea and faba bean protein isolates possesses inferior stability against lipid oxidation and physical stability than whey protein as fish oil emulsion stabilizer [99]. The soy protein isolate reduced the extent of lipid oxidation during the in vitro digestion of slightly oxidized sunflower and flaxseed oil [90]. The incorporation of proteins in the lipid-based system reduced the extent of lipid oxidation and generation of oxidation compounds (conjugated dienes in chains having also hydroperoxy/hydroxy groups, epoxides and aldehydes) [90].

*3.5. Effect of Protein Supplementation on Sugar Digestibility*

The intramolecular cross-linking of protein can occur during food processing, leading to molecular polymerization and covalent aggregations. Although this can reduce bacterial load, extend shelf life, and modify technological properties [101], they can negatively affect the nutritional value of proteins, depending on the processing conditions and the matrix of the ingredients or the diet [102]. The addition of sugar seemed to inhibit the crosslinks between amino acids (lysinoalanine, lanthionine) [102].

During industrial processing, prolonged storage, or in domestic cooking, the Maillard carbonyl-amine reaction is one of the non-toxic reactions that induce chemical modifications, creating color, aroma, texture, and other specific properties of foodstuffs [103]. Although the formation of these products can positively affect the sensory and technological properties of foods, it can induce the destruction of essential amino acids and the production of some anti-nutritive compounds [104]. To evaluate the effect of the generated protein–sugar association on the nutritional quality of the food, ref. [102] showed the impact of processing temperature and sugar type (glucose, xylose) on the extent and rate of soybean protein hydrolysis was investigated [102]. The effect of sugar addition on the extent and rate of proteolysis seemed to depend on the processing temperature. At relatively mild processing conditions (autoclaving at 100 °C), the effect of changes in the physical structure of proteins (protein aggregation) on the hydrolytic parameters looked at least as large as the effects of chemical changes to the amino acids. In contrast, under harsher processing conditions (120 °C), the chemical changes to the amino acids were more significant [102]. These changes would result to a higher amount of advanced Maillard reaction products, and a reduced lysine content [102]. The same effect was observed during Maillard reaction after the association of soybean protein isolate with D-galactose [105]. The lysine and arginine of soybean protein isolate covalently bonded to the carbonyl group of saccharides which reduced the lysine and arginine content [105]. These results were confirmed by the use of Maillard reaction to achieve high grafting degree during the conjugation between protein and polysaccharide [106]. Lysine and arginine residues formed covalent linkage between soybean protein isolate and maltodextrin or gum acacia, which decreased the above mentioned amino acids content [106]. Concerning furosine, which is generated at the early stage of the Maillard reaction and is considered a marker of the impairment of lysine residues in the protein [107,108], it was found that a 5 min extension of heat treatment (180 °C) could lead to a reduction of 60% in the furosine content [108].

On the other hand, the cross-linking of proteins and sugars of protein-rich food would generate melanoproteins [109]. The evaluation of the antioxidant activity of melanoproteins (100.19 mmol Trolox/kg) formed in a pea protein isolate/glucose model system during heating (180 °C for 5 min) was threefold higher than that of the initial pea protein [108]. In contrast, a more severe treatment (180 °C, 10 min) gave a lesser antioxidant capacity, which

can be related to a higher amount of insoluble melanoproteins. This fraction may remain in the gastrointestinal tract for a longer time and, may help in quenching the soluble radicals that are continuously formed in the intestinal tract, and possibly involved in the etiology of colon cancer [108].

The soybean protein isolate (SPI) and the *Pleurotus eryngii* polysaccharide (PEP) conjugate improved bioavailability of β-carotene in a simulated gastrointestinal tract, and reduced tert-butyl hydroperoxide-induced oxidative stress, thereby enhancing the antioxidant enzyme activities in Caco-2 cells [110]. The formation of bond between the soybean protein isolate with hydroxyl group of D-galactose augmented their antioxidant, antibacterial activity and the hypoglycemic effect [105].The use of soybean protein isolate-dextran conjugate to encapsulate curcumin to form nanoparticles revealed an enhancement of the antioxidant capacity that become more than double as compared to the curcumin alone [111]. These studies involving the interaction between the protein–polysaccharide complex/conjugate as delivery systems for bioactive ingredients looks rather challenging and holds the promise of more efficient delivery systems that could be uploaded with macro/micronutrients including other plant derived compounds, and to fortify the health status of individuals.

## 4. Conclusions

Nowadays, plant-based proteins present a promising solution to meet people's nutritional needs, and mitigate the challenges related to the increase in global population and environmental sustainability. As part of food development, many research studied the changes in functional and nutritional changes of protein isolate/concentrate as separate entity by assessing the possible effects of different treatments that can occur during food processing (i.e., cooking, high pressure, and irradiation). However, the coexistence of this protein with other components (i.e., carbohydrates, fats, polyphenols) requires an understanding of their possible interactions too.

Most of the studies showed that polyphenols can hamper protein digestibility, however, complexing of protein with appropriate phenol component can develop a functional ingredient with superior bioactivities. Similarly, starch and protein can affect each other's digestibility. Besides, addition of protein to the diet seems to reduce the negative effects of lipid accumulation in in vivo system. The type and concentration of biomolecules and other properties of food matrix play a crucial role in these interactions. Considered as the most reactive components, proteins can combine with the components of food system, inducing changes in the rheological and nutritional properties, including food behavior during oral processing and gastrointestinal digestion. Therefore, the optimization of processing conditions, such as temperature and pH, the selection of suitable combination of protein source and the food matrix, and concentration and type of biomolecules, can be exploited to develop a super food with improved nutritional and functional properties. Since the food is a complex system and its processing is a complex process, further detailed studies are needed, focusing on the effect of combined processing on the complexity of food system, with special regards on the functional, nutritional, and sensory properties of the final product.

**Author Contributions:** Conceptualization, P.S., C.N. and E.M.; writing—original draft preparation, C.N. and J.S.; writing—review and editing, P.S. and E.M.; supervision, P.S. All authors have read and agreed to the published version of the manuscript.

**Funding:** This research received no external funding.

**Institutional Review Board Statement:** Not applicable.

**Informed Consent Statement:** Not applicable.

**Data Availability Statement:** Not applicable.

**Conflicts of Interest:** The authors declare no conflict of interest.

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
