# Peer review of "Dough Rheological Properties and Macronutrient Bioavailability of Cereal Products Fortified through Legume Proteins"

_processes, doi:10.3390/pr11020417_

Round 1

Reviewer 1 Report

The topic of this manuscript is "Legume protein extracts: Impact on the rheological properties and nutritional value of bakery products", which should focus on the review on the connection between legume protein extracts and bakery products. However, quite a few paragraphs in this manuscript seem to have little relevance to the topic, and the logic are not clear enough. For instance, noodles are not bakery products which was not suitable for discussion in this manuscript, and the section from “3.3 Effect of polyphenols on protein digestibility” to the last section doesn't have much to do with the rheological properties and nutritional value of bakery products. In addition, the discussion and summary are not deep enough. Hence, I regret that I do not think it is of sufficient quality to be published at present.

Author Response

We would like to thank the reviewer for the careful reading of this manuscript and for the constructive suggestions.

We agree the comment that the noodles and pasta are out of the scope of the title, so we increased the product range concerned to cereal products.

The addition of legume protein concentrate/isolate aims to improve the quality of the protein-deficient products. For cereal products that are deficient in lysine, the supplementation with legume protein isolate will induce numerous changes not only in the nutritional value, but in the food rheology, affecting the final product's sensory quality too. In addition, during food processing, with the mixing of the food components (mainly: starch and fat) with protein, under specific conditions, many reactions can occur and affect the nutritional quality. In addition to these, cereal products often contain fruit ingredients, so the polyphenols they contain can also appear in the matrix and affect the protein utilisation. For this reason, we found important to examine the interactions of legume proteins with cereal proteins, carbohydrates and lipids, in addition to the with polyphenols, for which, for example, the effects of protein isolate and blueberry, grape and cranberry dosing were mentioned in section 3.3. 

The aim of the mentioned parts (3.3), (3.4), and (3.5) is for summarizing the effect of the addition of legume protein isolate on the nutritional quality. In general, for plant-based food, the nutritional quality depends on the amount of proteins, the distribution of amino acids, and, most importantly, the bioavailability of amino acids, which is determined by the intestinal absorption or digestibility. For this reason, in parts 3.3 and 3.5, the effect of present polyphenols and sugar on protein digestibility was studied. The assessment of the addition of legume protein on blood glucose and total cholesterol, which are among the cardiovascular and metabolic risk factors, can give an overview of the direct effect on health. In addition, the impact of the interaction of proteins and polyphenols, lipids, sugars and their potential in the amplification of the original effect of the compound (such as the antioxidant property of polyphenols or protein), the inhibition of reaction (such as lipid oxidation) or the generation of new compounds such as the Maillard reaction was also included in these sections.

Based on the review, we also carried out a thorough English language check and correction of the article. The changes are indicated in the manuscript with track change function of Word.

Reviewer 2 Report

Greetings and Regards

The main research question is Legume protein extracts: Impact on the rheological properties and nutritional value of bakery products.

The introduction is well described.

And considering the importance of process optimization and quality assessment in food industry, this article is suitable for publication in Food Processes section.

The results of the paper, according to the authors, attempt to show the interaction of legume proteins with grain nutrients, such as proteins, carbohydrates, lipids and polyphenols, and its effect on their bioavailability.

Figure 2 is not clear and needs a more comprehensive explanation.

In terms of written language, it needs some changes.

In general, the article is suitable for publication with a few changes

Author Response

We would like to thank the reviewer for the careful reading of this manuscript and for the constructive suggestions which helped to improve the quality of this manuscript.

Figure 2 is an assembly of the results of 4 scientific articles, partially discussed in the previous paragraph, which allows the perception of the effect of the supplementation of legume protein on G’, G’’ and tanδ of the dough/batter. Therefore, we added a summarizing section in the end of this paragraph to explain the obtained results better.

Based on the review, we also carried out a thorough English language check and correction of the article. The changes are indicated in the manuscript with track change function of Word.

Round 2

Reviewer 1 Report

The revised manuscript has been improved. However, the title should to be reconsiderated and the context need to re-organized again, cause the first section reviewed the rheological properties of doughs instead of cereal products, while the second section reviewed the effects or interaction effects of legume proteins on nutrient bioavailability in cereal products. 

Author Response

We would like to thank the reviewer for finding our corrections as appropriate. Indeed, the title remained misleading in the resubmitted version, as the rheological properties are understood to refer to the finished product and not to the dough. We have modified it now again and we believe that it adequately indicates that the rheological properties of the dough are discussed in this review by the nutritional effects.
Putting into the right context, we have modified the end of the introduction that sets out the purpose of the article (lines 75-80, if all track changes are visible). With this addition, we believe that we have now clarified that the rheological changes are not meant for the end products, but for the dough during the technological process.